# Cytotoxicity Evaluation and Antioxidant Activity of a Novel Drink Based on Roasted Avocado Seed Powder

**DOI:** 10.3390/plants11081083

**Published:** 2022-04-15

**Authors:** Andreea Pușcaș, Anda E. Tanislav, Romina A. Marc, Vlad Mureșan, Andruța E. Mureșan, Emoke Pall, Constantin Cerbu

**Affiliations:** 1Food Engineering Department, Faculty of Food Science and Technology, University of Agricultural Sciences and Veterinary Medicine Cluj-Napoca, 3-5 Calea Mănăştur Street, 400372 Cluj-Napoca, Romania; andreea.puscas@usamvcluj.ro (A.P.); anda.tanislav@usamvcluj.ro (A.E.T.); romina.vlaic@usamvcluj.ro (R.A.M.); vlad.muresan@usamvcluj.ro (V.M.); 2Department of Infectious Diseases, Faculty of Veterinary Medicine, University of Agricultural Sciences and Veterinary Medicine Cluj-Napoca, 3-5 Calea Mănăştur Street, 400372 Cluj-Napoca, Romania; emoke.pall@usamvcluj.ro (E.P.); constantin.cerbu@usamvcluj.ro (C.C.)

**Keywords:** avocado seed, valorization, coffee alternative, food waste, bioactive compounds, antioxidant capacity, cytotoxicity

## Abstract

The avocado seed is an underused waste resulting from the processing of pulp. Polyphenols, fibers, and carotenoids are present in the seed, which also exhibits prophylactic, fungicidal, and larvicidal effects. Developing food products with avocado seed as an ingredient or spice is highly desired for nutritional, environmental, and economic reasons. The present study proposed its valorization in a hot drink, similar to already existing coffee alternatives, obtained by infusing the roasted and grinded avocado seed. The proximate composition of the raw or conditioned avocado seed and that of the novel drink were determined. The total phenolic content was assessed using the Folin-Ciocâlteu method. The total carotenoids were extracted and assessed spectrophotometrically. Starch determination was performed by the Ewers Polarimetric method. The highest content of polyphenols, 772.90 mg GAE/100 g, was determined in the crude seed, while in the drink was as low as 17.55 mg GAE/100 g. However, the proposed drink demonstrated high antioxidant capacity, evaluated through the DPPH method. This might be due to the high content of the total carotenoid compounds determined in the roasted seed (6534.48 µg/100 g). The proposed drink demonstrated high antiproliferative activity on Hs27 and DLD-1 cell lines.

## 1. Introduction

The peel and seed of the avocado, resulting as by-products in the processing of the pulp, are waste materials which should have application in the domains of food, nutrition and medicine, since recent studies revealed and characterized the valuable nutrients they contain, demonstrating health promoting effects of some of their extracts [1,2,3,4]. The peel is edible, but is not consumed due to its bitter taste, chewiness, and because it is hard to digest. The seed is tough, has an astringent taste, and needs processing prior to consumption; therefore, formulating food products containing these valuable ingredients is challenging.

The pulp of avocado is a great source of proteins and monounsaturated fatty acids (predominantly oleic acid—62.14%, palmitic 17.2%, linoleic 11.11%, and palmitoleic acid 7.34%) and low amounts of stearic acid, 0.63%, protecting consumers against coronary heart disease development [5,6]. Besides fatty acids, the nutritional value of avocado is also due to antioxidants, carotenoids, phytochemicals such as α-tocopherol and β-sitosterol, and vitamins (B6, biotin, folic acid, thiamine, riboflavin, vitamin D and K) [7,8]. Avocado pulp has been used (fresh or dehydrated, defatted) in supplementing food products such as meat alternatives or in replacing wheat flour and butter, in whole grain crackers [9,10]. The processed products of avocado pulp include oil, the paste, puree, and guacamole; each of them is sensible to oxidation processes, so preservation or physico-chemical treatments are mandatory. It was reported that increased consumption levels can reduce adult weight gain [11], so it is no surprise avocado is becoming more and more consumed. Along with this, a high level of waste material is also generated on the industrial scale or in individual households.

The seed represents 13–18% of the weight of the whole fruit and the residues have a significant environmental impact due to the great organic charge they contain. Additionally, avocado seeds generate costs associated with disposing, handling, transport, and storage [7,12,13]. Numerous studies revealed some valuable compounds in the chemical composition of the avocado seed, imparting antioxidant and antimicrobial properties to it, transforming this waste material into a valuable ingredient of food products with the potential for medicinal use [3,14,15]. These studies were conducted on various species of avocado, such as Corillo [16] or Mill [17], including the Hass variety [18]. Phytosterols, triterpenes, fatty acids, furanoic acid, abscisic acid, proanthocyanidines (PACs), and other polyphenols are present in variable amounts in the seed, depending on the maturity, growth condition, and variety of the avocado [15,19,20].

Besides, anti-inflammatory, hypoglycemic, antihypertensive, fungicidal, larvicidal, hypolipidemic, analgesic, amoebicidal, and giardicidal activities of some extracts from the avocado seed have been reported [15,21,22,23,24,25]. The seed contains more soluble fibers than the pulp, so by ingesting it, one could naturally prevent constipation. It is also proved to be effective in the prophylaxis of gastric ulcers, preventing the occurrence of this disease [26,27]. Avocado seed is also rich in tannins, carotenoids, and tocopherols, which inhibited the in vitro growth of prostate cancer cell lines [28].

Extracts from the peel and seed displayed different functionalities when added to food products [17]. Acetone/water (70:10 *v/v*) extracts from the peel and seed were included in raw porcine patties and hindered the oxidative reactions and color deterioration during chilled storage of the product [29]. A natural orange colorant was extracted with water from the avocado seed which was priorly grinded [30].

Due to some antinutritional compounds present in the avocado seed, namely phytate, oxalate and cyanogenic glycosides [31], studies have been carried out in order to evaluate the effect of processing methods against the antinutritional compounds of *Persea americana* seed as a step towards establishing purposeful utilization in the food production area [13,32,33]. Solid-state fermentation of Hass avocado seed with *A. niger* GH1 led to an improved antioxidant activity [4]. Among other antinutritional factors, tannins, phytic acid and alkaloids were determined [1]. Soaking and boiling of the avocado seeds reduced the antinutritional compounds to a large extent. Some research describing novel food products with avocado seed as the ingredient has already been conducted [9,33,34,35]. Powders obtained from the seeds of avocado and peeled ginger roots were used for the preparation of eight prototypes of candies with sugar or aspartame. The candies were analyzed in terms of moisture content (77.3–92.5%), total sugars (0.10–0.66 mg/100 g) and microbial count. The sensorial analysis were promising, the candies being scored above average in taste, texture, and flavor [36].

Avocado seeds have been proposed to be valorized as flour and it would be of special use in tropical countries, were crops such as wheat, barley, millet, and rye do not lead to quality flours and lead to higher costs for pastry products [37]. For the flour preparation, the washed seeds were chopped and dried in an oven at 60 °C for 5 h. The seeds were then grounded in a pulverizer. The avocado seed flour was characterized in terms of yield, proximate analysis, gluten, and falling number and different biscuit formulations were prepared in order to test its applicability. The anthocyanin pigments of the avocado seed led to a darker color for the biscuits, which had an average acceptability in terms of the organoleptic properties. In conclusion, the avocado seed flour, despite the high nutritional value, would not be suitable for baking, having high values for the falling number and not forming the gluten network needed for structure [37]. However, Hass Avocado seed flour alone or in mixture with corn was studied for obtaining extruded snacks, showing promising results [35].

In the current study, drying and roasting were explored as conditioning treatments aiming to reduce the antinutritional compounds activity and concentration in the avocado seed. The present study proposed the use of the avocado seed powder in a hot drink, which could be consumed as a coffee alternative. Novel usages would increase its consumption and decrease food waste and pollution. The roasted avocado seed powder and thereof novel drink were analyzed, and the results revealed some antioxidant functionality and anti-carcinogenic effects.

## 2. Results

### 2.1. Optimization of Roasting of Avocado Seed and the Drink Preparation

Since the organoleptic properties of avocado seed were unknown, the temperature setting for the heat treatment was started at 135 °C, with a duration varying from 5–90 min. The best results in term of color and flavor were obtained after 90 min, when the seed was dark brown and well flavored. However, in order to reduce the duration of the roasting process, higher-temperature protocols were also explored. Table 1 presents the organoleptic profile of the avocado seed roasted under different time-temperatures protocols, starting from 160 °C ± 5 °C to 200 °C ± 5 °C. Given the results, the most suitable time-temperature protocol was: 180 °C/25 min, respectively 180 °C/30 min and 200 °C/5 min.

In order to obtain a product with optimal properties, different percentages of water and avocado powder were studied. The final composition was decided to be composed of 93% water and 7% roasted avocado seed powder. The powder had an intense and slightly astringent aroma, and it is sufficient even in this low percentage for obtaining the novel hot drink.

### 2.2. Proximate Composition

To determine the patterns of biologically active compounds accumulation in agro-industrial by-products, it is important to identify their composition and content in separate parts. The proximate composition of the Hess Avocado Seed, in raw, dried, or roasted state is summarized in Table 2. The results indicated that the seed is a good source of dietary protein, with 5.10 g/100 g Fresh Weight (FW) in the raw condition and 5.35 g/100 g FW when the seed is dried or 4.05 g/100 g FW sample when roasted at 160 °C for 60 min. The moisture was reduced in the drying or roasting treatments, leading to different proximate compositions of the samples. The raw avocado seed, besides protein, contains 0.74 g/100 g FW of fat and 49.72 g/100 g FW of carbohydrates and a content of 1.61 g/100 g FW minerals.

The raw avocado seed will furnish, upon consumption, 225.94 kcal/100 g FW and a higher caloric content was calculated for the dried and roasted samples (275.41/100 g and 395.37 kcal/100 g FW respectively). The currently proposed drink was prepared from 7% of the roasted avocado powder infused with hot water, and thus will have a total caloric amount of 56.48 kcal/100 mL.

The avocado seed is rich in carbohydrates, and thus the further investigation of starch content, which is the most predominant among the carbohydrates, was performed by the Ewers Polarimetric method. In the dried sample, the content is 43.9868 g/100 g FW and in the roasted sample is 48.1192 g/100 g FW. The thermal treatment of the sample might lead to some hydrolysis, increasing the content of starches in comparison with the dried sample. The rest carbohydrates might be represented by dietary fiber [38].

### 2.3. Total Polyphenol Content by Folin-Ciocâlteu Method

As can be seen in Table 3, the highest total polyphenol content was recorded in the crude avocado seed, with a value of 772.90 ± 4.09 mg GAE/100 g FW. The total amount of polyphenols was significantly reduced during processing, while in the drink was as low as 17.55 ± 0.70 mg GAE/100 g FW, given that only 7% of the roasted seed was used for the preparation of the drink. The total polyphenol content of the roasted avocado seed (180 °C/25 min) was of 179.07 ± 4.09 mg GAE/100 g FW; thus, the thermal treatment required in order to obtain organoleptic profiles similar to the coffee alternatives will significantly affect the biologically active compounds which are known to be thermolabile. However, polyphenols are generally more hydrophilic than lipophilic, and thus the proposed hot drink could be a good delivery system for these compounds.

### 2.4. The Total Carotenoid Compounds Content in the Avocado Seed

The total carotenoid compounds in fruits and vegetables are known to vary because of factors such as genetic variety, maturation stage, or processing conditions. The highest total carotenoid compound in the avocado seed was determined for the dried sample 9228.52 ± 21.20 µg/100 g FW. Similar, but statistically different contents, were determined for the crude and roasted samples (6190.56 ± 14.30 µg/100 g FW and 6534.48 ± 28.30 µg/100 g FW, respectively), as seen in Table 4.

### 2.5. Antioxidant Capacity of Avocado Seed and the Drink

The antioxidant capacity reflects the capacity of bioactive compounds to maintain the nutritional and sensorial quality of the product. The highest capacity was observed for the dried seed (95.66 ± 0.566 RSA% DPPH inhibited), but the results registered for the crude and roasted seed were not statistically different (Figure 1). On the other hand, good antioxidant capacity was registered for the proposed drink; even if the roasted avocado seed powder represents 7% of its composition, its antioxidant capacity was 90.27 RSA% DPPH inhibited.

### 2.6. The Total Acidity and pH Analysis

The organic acid content present in the crude avocado seed, determined as the titratable acidity, was 0.0538% malic acid equivalents, while in the dried and roasted samples the values were increased to 0.1361% malic acid equivalents and 0.1223% malic acid equivalents respectively. The novel drink registered a total acidity of 0.0268% malic acid equivalents. A pH value of 5.12 ± 0.13 was determined for the roasted avocado seed based drink, this being characteristic for acid drinks.

### 2.7. The Cytotoxicity Assay of Avocado Seed Drink

Cytotoxicity is considered an important aspect of any new food product or beverage which partially anticipates its health benefits upon consumption. Figure 2 and Figure 3 show the cell viability of the exposed cells to increasing concentrations of the novel roasted avocado seed-based drink.

In vitro cell viability tests demonstrated less than ≈6% loss in cell viability on both cell lines at the lowest concentration (2.5%) investigated. At the highest concentration (40%), loss in cell viability was observed to be ≈57% for Hs27 cells, whereas the value was ≈47% for the DLD-1 cells at the same concentration. These results showed that the novel drink may decrease the viability of both human fibroblasts (Hs27) and human colorectal adenocarcinoma (DLD-1) cell lines. Furthermore, cell viability experiments demonstrated a dose-dependent response on both cell lines.

## 3. Discussion

Dried Hass avocado seed powders (65 °C/120–180 min) obtained from ripe or unripe samples were analyzed in terms of proximate composition in another study, in order to be used for preparing extruded snacks [35]. The results of the proximate composition for the ripe seed were slightly different of those presented in the present study, probably due to the longer processing time: the moisture was lower—23.79%, 3.18% protein, 3.33% fibers, 65.62% nitrogen-free extract (carbohydrates), 2.6% ether extract, and 1.51% ashes. This study states the presence of antinutritional factors in the avocado seed (hydrocyanic acid, cyanogenic glycosides, condensed polyphenols and some tannins), which can be eliminated by a cooking treatment [35].

The proximate composition of avocado seed and that of the vitamins were determined in another study after boiling or soaking of the avocado seed. Statistically significant differences (*p* < 0.05) occurred for the content of crude fat, minerals, crude fiber, carbohydrates, and vitamins A, C, and E, during different processing protocols [32]. A content of 0.9% fat and 3.10% protein were determined by other authors in the crude (untreated) Hass Avocado seed [39]. The analysis of individual minerals with the atomic absorption spectrometry method was also performed, revealing high amounts of phosphorus (1000 mg/kg), calcium (533 mg/kg), and magnesium (544 mg/kg); an amount of 1.97% ash was determined, slightly higher than the amount determined in our study [39]. In the Algarvian avocado var. “Hass”, lower percentages of protein and a higher amount of fat were determined and the acidity of 2.67 ± 0.17 mg of tartaric acid equivalents/100 g [37]. The starch content of the avocado seed was also investigated by other authors with the Ewers polarimetric method, and the study revealed that higher amounts of starch can be determined in the ripe seed than in the over-ripe seed [39]. Another study reported a starch yield of 42.2% extracted with metabisulfite solution and by producing a dough which was filtered and washed to separate starch from *Daisy* variety [40]. This starches can be further hydrolized using acid or enzyme hydrolysis [41]. The parameters of gelatinization and viscosity of extracted from the avocado seed present restricted dilation, which suggests their possible use in food products which must be heated up at 100 °C [35]. A comparative analysis of the antioxidant capacity of different varieties of avocado seed samples and different extraction condition was carried out in the study of Segovia et al. [42]. In regard to the analysis of total phenolic compounds, the following results were registered for the *Persea americana var.* Hass seed, extracted with Methanol/Water (80:20 *v/v*), 60 °C, namely 9.51 ± 0.16 mg GAE/g [42]. For the ethanolic extracts, the results were of 8.07 ± 0.03 mg GAE/g in the study of Amado et al. [8]. Similar total phenolic compounds were revealed in the Algarvian avocado var. “Hass” (7.04 ± 0.13 mg GAE/g) [43]. Thus, the extraction method applied in the present study is effective in the determination of the total phenolic compounds.

The determination of the total phenolic compounds in a Turkish chicory root, which was roasted for 2 h at 140 °C and grinded to be used as a common coffee alternative, revealed lower contents (between 0.943–13.860 mg GAE/g DW) than those determined in the roasted avocado seed (180 °C/25 min), which were 179.07 ± 4.09 mg GAE/100 g [44]. In the study of Afify et al., the TPC determined for different coffee or teas prepared in hot or cold water, varied between 1.68 ± 0.06 to 2.28 ± 0.06 g GAE/100 g in teas and 1.87 ± 0.07 for a coffee variety, which is lower than the amount determined for the drink prepared from the roasted avocado seed (17.55 ± 0.70 mg GAE/100 g) [45].

Given the high availability, economic advantages, and the abundance of the biologically active compounds, it would be of use to assess the acceptability of the consumers toward food products or beverages having avocado seed as ingredient or spice. It is also a good source of carotenoid compounds, a total amount of 0.97 ± 0.164 mg/100 g fresh weight basis expressed as β-carotene equivalents being determined for the Algarvian avocado var. “Hass”. This result is higher than the results exhibited by the crude sample explored in our study. In the current study, slightly higher amounts of total carotenoid compounds were detected in the roasted sample, in comparison with the raw sample, which might be caused by the increase in different isomers of lycopene which are precursors of β -carotene [46].

Another study explored the individual carotenoid compounds determined by HPLC-MS in the Hass Avocado variety originating from Chile, lutein (131.51 μg/g oil extracted from the seed or 2.62 μg/g fresh fresh weight basis) and *β*-carotene (111.88 μg/g oil extracted from the avocado seed or 2.22 μg/g fresh weight basis) being determined as major carotenoid compounds [47]. Numerous studies explored and demonstrated the in vitro antioxidant and cancer inhibitory activity of avocado seed extracts [2]. Cell viability was assessed using a modified MTT assay on the seed extracts of Hass and Fuerte varieties, and their capacity to inhibit TNFα was assessed in LPS-stimulated RAW 264.7 macrophage culture. None of the tested extracts exhibited cytotoxicity up to 10 μg/mL and the seed exhibited good anti-inflammatory effects after 4 h [48]. In another study, a methanol soluble fraction of the avocado displayed the capability to induce apoptosis and anti-proliferative effects to MCF-7 cell lines [49].

## 4. Materials and Methods

### 4.1. Materials

Hass Avocados were purchased from a local market from Cluj Napoca, Romania and were of Columbian origin. All the reagents were of analytical grade.

### 4.2. Optimization of Roasting of Avocado Seed

The roasting of the avocado seed was performed at different time and temperature intervals to highlight the aroma and to obtain organoleptic characteristics as close as possible to those of coffee or its replacements. The organoleptic analysis was conducted by analyzing the flavor profile of the seed after each time and temperature protocol by a part of the collective of authors who were instructed in regard to the desired organoleptic properties and with previous experience in this [50]. The roasting process was conducted in a Memmert UF55 (Buechenbach, Germany) oven (135 °C for 5–90 min first and for optimization different time temperatures intervals between 160–200 °C, 5–90 min were explored). The temperature in the room was 22–23 °C and the relative humidity was 40–42%.

### 4.3. The Avocado Seed Powder and the Drink Preparation

The fresh avocado seed contains a high amount of water in the composition and the seeds were naturally dried in a warm airy room, for 5–7 days. The seed is a dicotyledonous and it was kept for 5 min in the oven to facilitate the decortication. The seed was passed through a grater prior to roasting at 180 °C for 25 min. The grater was of stainless steel grade and the side with small holes (diameter 2 mm) was used. After roasting, it was finely ground into a powder using the Retsch RM200 (Haan, Germany) grinding machine set in position 8 (100 rot/min for 20 min).

For the hot drink preparation, water (90 °C ± 5 °C) and a French press were involved until the infusion took place (10 min).

### 4.4. Proximate Composition Analysis

The chemical compositions including moisture, ash, total carbohydrates, total sugars, crude fat, and protein content were determined for the fresh, dried, and roasted avocado seeds according to AOAC procedures and were expressed in regard to the fresh weight (FW). For moisture analysis, samples were subjected to drying in an oven at 103 ± 2 °C for 3 h, the experiment being repeated until the weight was constant. The samples were cooled in a desiccator for one hour and weighed (AOAC, 1999).

The ash content was determined by calcination at 550 °C of 2 g of probe until a gray ash was obtained, with the removal of the carbon black spots by splashing with water, then the process was continued until a gray or white ash resulted (after 6 h).

The crude protein content of the samples was estimated by the Kjeldahl method.

The crude fat content of the samples was determined by extracting a known weight of powdered samples (3 g) with petroleum ether as a solvent, using the Soxhlet apparatus.

The amount of total carbohydrate was calculated by difference.

The starch content of avocado seed was determined using the Ewers polarimetric method (ISO 10520: 1997) with some modifications [51].

### 4.5. Total Phenolic Content

The total polyphenols content was assessed using the Folin–Ciocâlteu method [52], slightly modified. An amount of 1 g of sample was mixed with methanol and 0.01% HCl. The obtained extracts were filtered and dried at 35 °C under reduced pressure (Heidolph Rotary Evaporator, Schwabah, Germany). A quantity of 25 µL sample was mixed with 1.8 mL of distilled water and 120 µL Folin-Ciocâlteu reagent in a glass vial. A 7.5% Na_2_CO_3_ solution prepared in distilled water (340 µL) was added 5 min later to assure basic conditions (pH 10) for the Redox reaction between the phenolic compounds and the Folin-Ciocâlteu reagent. The samples were incubated for 90 min at room temperature. Methanol was used as a control sample. The absorbance at 750 nm was measured using a Shimadzu UV-VIS 1700 spectrophotometer (Shimadzu, Kyoto, Japan). The calibration curve was plotted based on the 0.25, 0.50, 0.75, 1 mg ml^−1^ concentration of gallic acid. The total polyphenol content of the avocado seed was expressed for fresh weight (FW) in Gallic acid equivalents (GAE)—mg GAE·100 g^−1^.

### 4.6. Spectroscopic Analysis of the Total Carotenoid Content

To assess the total carotenoid content, carotenoids were extracted from the crude, dried, and roasted avocado seeds, using ethanol: ethyl acetate: petroleum ether (1:1:1, *v/v/v*). Successive extractions were performed. The extracts were combined, filtered, and washed with distilled water, diethyl ether, and a saturated solution of NaOH. The ethereal phase was recovered and subjected to rotary evaporation at 35 °C (Heidolph Rotary Evaporator, Schwabah, Germany). Estimation of carotenoids was spectrophotometrically determined using Shimadzu UV-VIS 1700 set at 450 nm (Shimadzu, Kyoto, Japan).

### 4.7. The Antioxidant Activity

The antioxidant activity was determined using the 2.2-diphenyl-1-picrylhydrazyl (DPPH) method, according to Mureșan et al. [53]. 10 µL methanolic extract from the avocado seed was mixed with 3.9 mL DPPH methanolic solution (0.025 g/L) and 90 µL distilled water. The mixtures were stirred and maintained properly in the dark for 30 min. The absorbance of the samples was measured at 515 nm (Shimadzu 1700 UVVIS, Kyoto, Japan) against a methanol blank. The positive control was prepared using a gallic acid solution (0.5 mg/mL). The negative control was prepared using methanol. Results were expressed as percent over standard DPPH absorbance according to the following equation:$$\mathrm{RSA}\;\left[\%\right]=\frac{A_{\mathrm{DPPH}}-{\;A}_{P}\;}{A_{\mathrm{DPPH}}}\;•100$$
where: RSA [%]—Radical Scavenging Activity; A_DPPH_—the absorbance of DPPH solution with methanol; A_P_—the absorbance of DPPH solution after 30 min incubation with sample.

### 4.8. The Total Acidity and pH Analysis

The total acidity was performed by neutralization with sodium hydroxide solution (0.1 N) in the presence of fenolftalein as indicator. The results were expressed in g malic acid equivalents/100 g. Titratable acidity calculation was carried out using the formula:$$\mathrm{Acidity}\;\%\;(\mathrm{malic}\;\mathrm{acid})=\frac{V\;•\;0.0067}{m}\;•\;100,\;$$
where ‘V’ is the volume of NaOH solution 0.1N, ‘m’ is the weight of the sample, and ‘0.0067′ g of malic acid corresponds to 1 mL NaOH 0.1 N.

pH analysis was also carried out on the novel drink. The determination was based on the property of indicators contained by the pH paper to change their color in the presence of hydrogen ions. To determine the pH of the novel drink, the pH meter Mettler Toledo (Columbus, OH, USA) was employed.

### 4.9. Cytotoxicity Assay

The cytotoxicity assay of the avocado seed drink was performed using human fibroblasts Hs27 (ATCC^®^ CRL-1634^™^, Manassas, VA, USA) and human colorectal adenocarcinoma DLD-1 (ATCC^®^ CCL-221^™^, Manassas, VA, USA) cell lines. The cells were cultured according to standard conditions. The potential cytotoxicity of avocado seed drink was assessed with (4,5-dimethylthiazol-2-yl)-2,5-diphenyltetrazolium bromide (MTT) assay. In order to obtain cell suspensions, the cells were treated with 0.25% trypsin-EDTA, and after centrifugation (1500 rpm for 5 min), 1 × 10^4^ cells/well were seeded on 96-well plates in 200 µL complete culture medium. After 24 h, 80 µL (40% *v/v*), 60 µL (30% *v/v*), 40 µL (20% *v/v*), 10 µL (5% *v/v*), and 5 µL (2.5% *v/v*) of avocado seed drink were added, while removing the same volume of culture media, resulting in a total final volume of 200 µL/well. Control samples were represented by untreated cells. Each experimental condition was performed in triplicate. Cell proliferation analysis was performed after 24 h. After 24 h, the medium was removed and 100 µL of 1 mg/mL MTT solution (Sigma-Aldrich, St. Louis, MO, USA) was added. After 4 h of incubation at 37 °C in dark, the MTT solution was removed from each well and 150 µL of DMSO (dimethyl sulfoxide) solution (Fluka, Buchs, Switzerland) was added. Spectrophotometric readings at 450 nm were performed with a BioTek Synergy 2 microplate reader (Winooski, VT, USA). Data are shown as percentage of cell viability.

### 4.10. Statistics

Statistical differences were obtained through an analysis of variance (ANOVA) followed by Tukey’s multiple comparison test at 95% confidence level (*p* ≤ 0.05).

## 5. Conclusions

The avocado seed has been explored lately for various applications due to its composition rich in bioactive compounds. Numerous processing techniques such as boiling, dryingor extrusion, have been explored so far as conditioning methods, due to the presence of anti-nutritive factors in its composition. The current study evaluated different time-temperature protocols for the roasting of the avocado seed, along with drying, and the flavor and color modification were assessed. We proposed the valorization of the roasted (180 °C/25 min) avocado seed powder in a hot drink, obtained by creating an infusion with 7% of the powder and hot water.

The raw or conditioned (dried or roasted) Hass avocado seeds were examined in terms of proximate composition and bioactive compounds. The seed possess a high amount of carbohydrates, including dietary fibers and between 4–5% protein, depending on the conditioning process applied (drying or roasting). The total polyphenolic content of the avocado seed was reduced during the conditioning, while the acidity and total carotenoid compound were significantly increased. The novel drink exhibited a high antioxidant capacity of 90.27 RSA% DPPH inhibited, which might be due to the presence of carotenoid compounds or flavonoids.

However, the novel drink exhibited a lower concentration of the total polyphenolic compounds in comparison with the raw or conditioned seed (only 17.55 ± 0.70 mg GAE/100 g in the drink compared with 179.07 ± 4.09 mg GAE/100 g in the roasted avocado seed), mostly because the drink has a roasted avocado seed powder concentration as low as 7%. This is a higher content that what was previously registered for coffee or coffee surrogates in other studies. The cytotoxic properties of the novel drink based on roasted avocado seed were also demonstrated, because when it was applied in a concentration of 40% on DLD-1 cells and Hs27 cells during the MTT cytotoxicity assay, it affected the viability of the cells.

## Figures and Tables

**Figure 1 plants-11-01083-f001:**
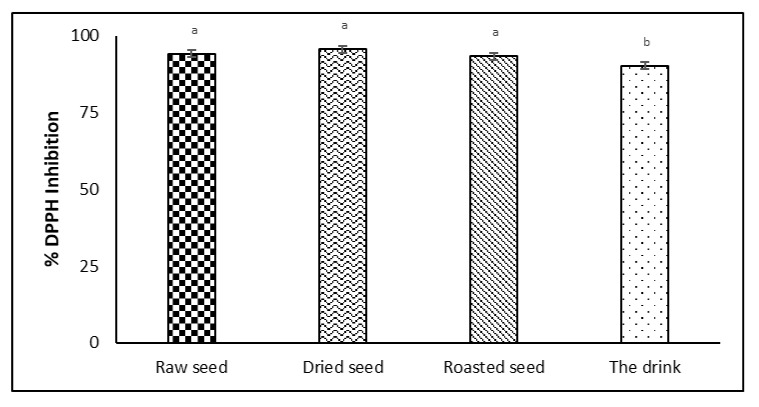
Antioxidant capacity as measured through DPPH assay; identical superscript letters indicate no significant difference (*p* > 0.05).

**Figure 2 plants-11-01083-f002:**
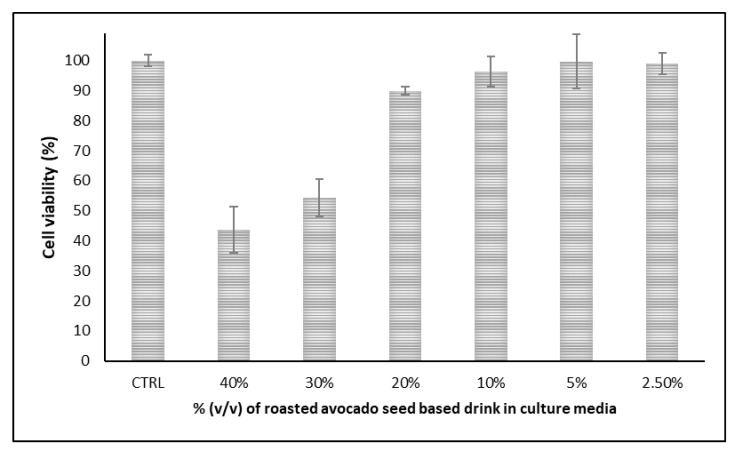
Cell viability of Hs27 cells when treated with the novel roasted avocado seed-based drink.

**Figure 3 plants-11-01083-f003:**
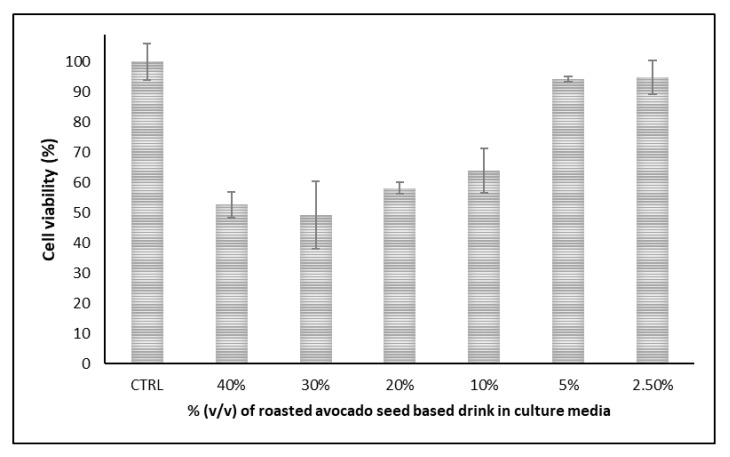
Cell viability of DLD-1 cells when treated with the novel roasted avocado seed-based drink.

**Table 1 plants-11-01083-t001:** Organoleptic profile of roasted avocado seed at different temperature-time intervals.

Time [min]	Temperature (°C)					
	160 °C ± 5 °C		180 °C ± 5 °C		200 °C	
	Color	Flavor	Color	Flavor	Color	Flavor
5	unchanged		unchanged	unchanged	dark brown	very flavored
10	light orange	no flavor		slight flavor	dark brown	carbonized
15			light brown	slight flavor	black	
20	dark orange	no flavor	slight dark brown	flavored		
30	light brown	slight flavor	dark brown	well flavored		
35		flavored	very dark brown	carbonized		
45	slight dark brown					
60	dark brown	well flavored				
75	very dark brown	strong flavor				
90	very dark brown	carbonized				

**Table 2 plants-11-01083-t002:** Proximate composition of raw or conditioned Hass avocado seed expressed as g/100 g (FW).

Sample	Moisture	Protein	Fat	Carbohydrates	Minerals
	g/100 g FW				
Raw	42.83 ± 2.26	5.10 ± 0.003	0.74 ± 0.003	49.72 ± 2.27	1.61 ± 0.01
Dried	29.25 ± 0.16	5.35 ± 0.12	0.65 ± 0.06	62.04 ± 0.18	2.71 ± 0.15
Roasted	0.80 ± 0.01	4.05 ± 0.33	2.12 ± 0.13	90.02 ± 0.41	3.01 ± 0.07

**Table 3 plants-11-01083-t003:** The total polyphenol content of raw and conditioned Hass avocado seed and the novel drink.

Samples	Total Phenolic Content (mg GAE/100 g FW)
Crude seed	772.90 ^a^ ± 4.09
Dried seed	279.84 ^b^ ± 2.87
Roasted seed	179.07 ^c^ ± 4.09
The drink	17.55 ^d^ ± 0.70

Identical superscript letters indicate no significant difference (*p* > 0.05).

**Table 4 plants-11-01083-t004:** The total carotenoid compounds of raw and conditioned Hass avocado seed.

Samples	Total Carotenoid Compounds (µg/100 g FW)
Crude seed	6190.56 ^a^ ± 14.30
Dried seed	9228.52 ^b^ ± 21.20
Roasted seed	6534.48 ^c^ ± 28.30

Identical superscript letters indicate no significant difference (*p* > 0.05).

## Data Availability

Not applicable.

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
