# Peer review of "Cytotoxicity Evaluation and Antioxidant Activity of a Novel Drink Based on Roasted Avocado Seed Powder"

_plants, 2022, doi:10.3390/plants11081083_

Round 1
Reviewer 1 Report
The manuscript presents information about the possibility of uses of avocado seeds powder for drink formulation. In this manuscript antioxidant activity, phenolic content, cytotoxicity and proximate composition of avocado powder or/and drink were evaluated. The idea of this research is interesting, but in my opinion the paper needs some improvement and clarification before publication.
General comments
I think the title of Manuscript should be changed to better reflect content. The authors evaluated the cytotoxicity of drink prepared from avocado seeds powder, not from seeds powder itself. Maybe rearrange your results, first describe raw, dried and roasted avocado seeds, then present results of the obtained drink. Also the conclusion section should be improved. Some sentences for me are not conclusions or not based on obtained results, for example “organoleptic properties of avocado seed during roasting under various time- temperature protocols were described” or “The novel drink has an energetic value of 56.4823 kcal/100 ml”.
You provided organoleptic profile in table 1, but in the method section you did not mention anything about sensory analysis. How was it conducted?
Unify description of material, for example Table 2 – Raw, Table 3 crude seed, hot drink, Figure 1 crude seed, end product.
Also results should be rounded, for example, what is accuracy of data presented in table 2. In table 4 in error you provide one digit after decimal point, but in results two digits.
Specific comments
Line 15 What does it mean “prophylactic effect”?
Line 24 What is the unit of polyphenols content?
Line 26 To many significant digits, results should be rounded.
“The proposed drink is safe to be consumed…” More research is needed to assess the safety of seed drink.
Line 27 This product demonstrated high antiproliferative activity? Did you measure antiproliferative activity or cell viability?
Line 37 “flat taste” colloquialism.
Line 39 – 40 fatty acids composition, provide percentage of all.
Line 41 I am not convinced that these fatty acids protect but just could reduce risk of coronary heart diseases.
Line 42-48 References needed.
Line 49 “can reduce adult weight gain” - what does it mean? Could reduce weight or protect against obesity?
Line 80 Rather “antinutritional compound” than “antinutritional factors”.
Line 82 – 89 Paragraph is too long, too many details, please shorten it.
Line 94 Too many details.
Line 97 – 99 Anthocyanins are red and easily decomposed during heating. What about Maillard reaction compounds?
Line 117 - 118 This sentence is confusing. Also you mentioned 25 minutes roasting time, but is not supported by results presented in table 1.
Line 122 Why did you assume 7% of avocado powder addition?
Line 126 Table caption – where is 135°C, in table are 160, 180 and 200°C.
Correct mistyping in table, for example “falvored”.
Line 135 – 136 You mixed units g/100 g and percent. Use the same units. Is it calculated based on fresh or dry matter?
Line 137, Table 2 What is the unit of fat content? Why did fat content increase after roasting? Maybe calculation based on dry matter could better reflect changes?
Line 164, table 3 Results are expressed based on fresh or dry matter? In case of drink was per g or ml?
Line 187 Remove green background of chart, adjust scale, maybe from 0 to 100. What is it “End product”? Is it a drink? Unify nomenclature.
Line 194 pH value is rounded?
Line 196 Did you test antiproliferative activity or cell viability?
Line 249 Compared results were obtained from avocado flesh or seed? Express it in the same unit (mg/100g maybe).
Line 258 Again, flesh, seed or all fruit?
Line 310 Antioxidant activity is expressed as %RSA. How did you calculate this parameter? Provide formulas.
Line 312 Why did you use capital letters?
Line 319 You could provide pH results if you conducted this measurement by pH-meter. In my opinion pH paper strips are suitable only for rough measurement.
Line 322 Include roasting in title. Provide more details about roasting, you used different temperatures. What about sensory analysis of roasted avocado seeds?
Line 326 Provide more details about drink preparation, at least water temperature and time. It is a very important step for final product properties.
Line 356 – 357 High antioxidant, but low amount of antioxidant compounds? You did not analyze the carotenoid content in drink.
Line 363 – 365 What is the conclusion in this sentence?
Reviewer 2 Report
The manuscript describes the cytotoxicity evaluation and antioxidant activity of avocado seed powder. The article is reasonably well organized, and it touches on a crucial issue, waste reduction. Authors are contributing to a noble cause. Unfortunately, I believe that the article merits publication for other journals focused on food chemistry. "Plants" is not a good fit, particularly not in "Spicy and Aromatic Plants" special issue. I encourage authors to submit other journals if Plants cannot accept it.
Major concern.
Findings are quite basic, and the big majority of the results regarding the composition (and biological activity in some systems) of the avocado seed have been published elsewhere with plenty of detail (https://doi.org/10.1016/j.sjbs.2021.02.087, https://doi.org/10.1016/j.fbp.2020.10.012, https://doi.org/10.1016/j.foodchem.2021.131469, https://doi.org/10.1016/j.foodres.2017.11.082, https://doi.org/10.1016/j.foodres.2017.11.082, etc).
Some minor concerns.
1- Some statements can produce controversy without solid references, ie. "protecting consumers against coronary heart diseases". This is debatable when working with natural products that vary along the production line. Please provide valid references for this.
2-How was caloric content evaluated?
3- There are no significant figures in the pH of 5. What is the precision of this number? How was measured?
4-When the total polyphenol content of the roasted avocado seed is discussed, please put it in a meaningful context, e.g. how this compares with other important commodities such as coffee.
5-Plots require better quality in the format but also in the content. So, for instance, figures 2 and 3 don't show SE or relative standard deviation.
Line 247- Hass seed, extracted >> no italics for "Hass"
Reviewer 3 Report
I found your article very interesting, but I have a few observations to make:
- Why in table 1 some timelines do not have both color and flavor described?
- Figure 1 could be made more similar to Figures 2 and 3 to be more understandable
- Figure 1 and Table 1 are not centered in the page
- In section 4.5 the DDPH method was mentioned but is not described
- In the bibliography the reference 29 is not homogeneous with the others
Reviewer 4 Report
Overall, I found the paper quite interesting. I had no idea there was a potential use for avocado seeds as a coffee substitute. In general, I believe this manuscript why of publishing, however, there are multitudinous grammatical errors contained throughout the paper.
To specific comments:
- the first line in the introduction directly contrasts what was stated in the first line of the abstract. Please amend.
- please insert ref at the end of the sentence as follows:
- lines 41, 46, 55, 83 (requires more than one ref), 85, 94 and 217.
- in line 47, please replace the word "sensible" with "sensitive."
- line 73, the statement "extracted from the avocado seed crushed with water" does not make sense. Please amend.
- line 76, you state in the sentence that some studies i.e. plural, have been performed to evaluate the effect of processing methods, but you have only referenced a single study. Please include other relevant studies, or if their is only the one study, then please amend the text to reflect this.
- lines 149 and 223, please insert space between last word and reference.
- line 173, from the word "slightly" to the end of the sentence (line 175), this part of the sentence does not belong in the results. You must only ever present pure results, not discuss their potential meaning/significance. Hence, please move the sentence to the relevant section in your discussion.
- Figures 2 and 3, where are the error bars on each graph? please insert into both figure. Also, on the x axis, "2,50%" should be written as 2.5% in both graphs.
- I am curious as to the future directions of this research. please include in the discussion.
- With regards to the methods:
- were the Hass avocados properly authenticated? If so, where and by whom? Please include details.
- line 326, what type of grater was used? Please include details.
- line 327, how was the roasted and grated avocado see ground into a powder? Please include details.
Round 2
Reviewer 1 Report
The authors have done a good job revising the document and incorporating my suggestions in the revised manuscript. Unfortunately, there seems to be still something to improve.
General
Please read your manuscript carefully, there are many mistyping. Improve some statements to be less confusing and much consistent.
Please read some tips how to rounding scientific results. Maybe https://www.jtcvs.org/article/S0022-5223(16)31101-1/fulltext#relatedArticles and https://www.ncbi.nlm.nih.gov/pmc/articles/PMC4483789/ could be helpful. Then round your results in whole manuscript.
Specific comments
Line 42 Provide percentage of undesirable stearic acid too.
Line 85-87 I suggest to incorporate these statement into objectives presented at the end of Introduction section.
Line 125, Table 1. In my opinion caption still not reflects content (aption: Organoleptic profile of roasted avocado seed at 135 °C and 160 °C at different time intervals - table presents temperatures 160, 180 and 200°C).
There is still time 20 min in table and mistyping “fflovored”.
Line 126-129 Remove this sentence (or move to Method section).
Line 147, Table 2 Rewrite: Table 2. Proximate composition of raw or conditioned Hass avocado seed expressed as g/100 g FW.
Line 164, 166, 168 Remove superscript from “772.90c ± 4.09 mg GAE /100g FW”. Round values according rules.
Line 182, 183 Remove superscript from “9228.52c ± 21.20”. Round values according rules. Change unit from mg/100 g to mg/100 g.
Line 195, Figure 1, Add OY axis title (% of DPPH inhibition). Remove legend, it is duplicate axis title.
Line 216-225, Figure 2, Figure 3 Remove digits after decimal separator in OY axis. OX axis title should be rather “% (v/v) of roasted avocado seed based drink in culture media” or something similar.
Line 236 Rewrite “The proxime and vitamins composition…”. How does boiling influence on vitamins? It was only affected on proxime composition?
Line 269 Improve style of reference.
Line 304 How many panelist conducted sensory analysis? What kind of sensory analysis was conducted – for example flavor profile method or Quantitative Descriptive Analysis?
Line 306-307 Room temperature and humidity are not important and necessary, provide conditions (at least time and temperature variations) of roasting.
Line 359 ADPPH should be in subscript. ADPPH is it absorbance of DPPH solution or DPPH solution with methanol (“negative control”)? AP is it sample absorbance or absorbance of DPPH solution after 30 minutes incubation with sample?
Line 371-373 Rewrite this sentence: “The determination was also conducted on a pH meter Mettler Toledo.”
Line 375-382, point 4.8 Move this point “4.8 The avocado seed powder and the drink preparation” after point “4.2 Optimization of roasting of avocado seed”.
Line 380 What was “small wholes”? Provide diameter.
Line 390-391 Plates has 200 ml volume. You add some volumes of avocado seeds drink. What solution you used to fill up to volume? Improve description. Recalculate percentage of addition of avocado seed drink “2.5 μl (2.5%)”.
Line 409 Side effect of roasting is removing some water, but is not drying.
Line 413 Again I am confused. Did you tested roasting or drying your samples?
Reviewer 2 Report
The authors completed a compelling response to my observations. I believe the manuscript can be considered for publication if Plants consider this topic is a good fit for the "Spicy and Aromatic Plants" special issue. Please do a final check for typos and grammatical errors. e.g L35 “demonstratinghealth” a space is missing, and so on.
